# Association between the use of Information and Communication Technology and activeness in life among community-dwelling older adults

Kanna Tezuka[1]*, Yachiyo Sasaki[2], Midori Shirai[1]

1 Graduate School of Nursing, Osaka Metropeolitan University, Osaka, Japan, 2 School of Health Science, Faculty of Medicine, Kagoshima University, Kagoshima, Japan

* so22611b@st.omu.ac.jp

## Abstract

This study aimed to clarify the association between the use of Information and Communication Technology (ICT) and activeness in life among community-dwelling older adults. A self-administered, unmarked questionnaire survey was conducted among individuals aged 60 years or older who were registered with Silver Human Resource Centers or Senior citizen clubs in a city in Osaka Prefecture. The survey collected data on participants, demographic characteristics, health status, ICT use, and activeness in life (particularly expansion of life space, exercise habits, motivation, and social activities). ICT use was defined as the use of any mobile devices to connect to the Internet, such as smartphones, tablets, or wearable devices. A logistic regression analysis was performed to estimate the association between ICT use and activeness in life after adjusting for confounding factors. A total of 892 responses were used in the analysis. The results revealed that the odds ratios (ORs) for expansion of life space (1.84) and motivation (2.17) were significantly higher for the group of participants using ICT use than for the group of participants not using ICT. In contrast, ICT use was not associated with exercise habits and social activities. Overall, this study clarified that ICT use is significantly associated with the expansion of life space and motivation among community-dwelling older adults. In particularly, ICT use may increase mental activity, inferred from the highest OR value for motivation noted in the current results. In conclusion, ICT use can enhance activeness in life among community-dwelling older adults.

**Data Availability Statement:** Ethical review at our institution is based on the Declaration of Helsinki and Ethical Guidelines for Medical and

## Introduction

The aging rate in Japan is 29.0% and is expected to continue to rise [1]; thus, a challenge emerges with regard to extending healthy life expectancy [2]. Community-dwelling older adults are expected to maintain and improve their life functions and physical and mental health through lifestyle modification [2]. To achieve this, it is necessary to support increased activeness in life among older adults.

Biological Research Involving Human Subjects provided by the Japanese Ministry of Education, Culture, Sports, Science and Technology and Ministry of Health, Labour and Welfare and Ministry of Economy, Trade and Industry. In these guidelines, questionnaire such as sex, affiliation, and medical history are deemed as personal information requiring special consideration. This study did not obtain personally identifiable data on human research participants such as name and birth dates. However, the participants in this study were a small group of individuals, Silver Human Resource Centers and senior citizen clubs, and the questionnaire included indirect identifiers such as sex, affiliation and medical history. Therefore, there is a risk the identification of study participants, and consent for data provision has not been obtained from the participants. For these reasons, there are restrictions to data sharing by the Research Ethics Review. The name of the institution is the Graduate School of Nursing, Osaka Metropolitan University Research Ethics Review committee. Adress: 1-5-17 Asahimachi, Abeno, Osaka, 545-0051, Japan URL: https://www.omu.ac.jp/nurs/en/ Phone: +81-6-6645-3511

**Funding:** The author(s) received no specific funding for this work.

**Competing interests:** No authors have competing interests.

In recent years, the use of Information and Communication Technology (ICT) has been promoted in various areas of Japanese society, including healthcare and regional revitalization [3]. The rate of Internet use among adults aged 70–79 years and 80 years or older is 65.5% and 33.2%, respectively, and these rates are increasing every year [4]. With such digitization of the society, various efforts are being made to promote ICT use, such as providing support to older adults in ICT use.

Recent studies have shown that ICT use is associated with higher self-rated health and happiness among older adults [5] and that a greater degree of ICT use promotes more social activities [6]. In a study conducted by the Ministry of Internal Affairs and Communications [7], older adults pointed out that ICT use helped bring about benefits such as increased communication, activity, enjoyment, and joy; improved health and a better sense of place and role; and increased motivation and life satisfaction. These findings suggest that ICT use may enable them to live more active lives.

Previous studies on maintaining and improving healthy life expectancy and life functions among older adults have evaluated physical factors such as the performance of activities of daily living [8], expansion of the life space [9], and engagement in physical activity [9, 10]; mental factors such as self-rated health and depression [11]; and social factors such as social participation and social support [11]. Thus, it can be inferred that one's activeness in life depends on whether or not one actively performs the actions necessary for life, and this can be evaluated based on three aspects: physical, mental, and social aspects.

However, few studies have examined ICT use and older adults' activeness in life based on all three aspects (i.e., physical, mental, and social aspects). Therefore, this study aimed to clarify the association between ICT use and activeness in life among community-dwelling older adults by evaluating the physical, mental, and social aspects of activeness in life.

## Materials and methods

### Participants

The study participants were adults aged 60 years or older who were registered with Silver Human Resource Center or senior citizen clubs in a city in Osaka Prefecture. Individuals aged 60 years or older who are interested in performing temporary short-term community-based work in exchange for remuneration can register with Silver Human Resource Centers. Meanwhile, senior citizen clubs resister individuals aged 60 years or older who are interested in engaging in voluntary community-based activities.

All the participants were handed a document describing the nature, aims, procedures, and possible risks of the study before their informed consent was obtained. No personal information was collected as part of the study. Consent was confirmed by ticking the consent box on the survey from. Furthermore, the study protocol was approved by the ethics committee of the Osaka Metropolitan University Graduate School of Nursing and the study was performed in accordance with the Declaration of Helsinki (2022–27).

### Data collection

A survey was conducted from June to July 2023 using an unmarked self-administered questionnaire. This questionnaire collected information on demographics (sex, age, living status [living alone or not], education level, and concerns about one's economic situation [presence or absence of concerns]); health status (medically diagnosed chronic disease and subjective symptoms); ICT use; and activeness in life (expansion of life space, exercise habits, motivation, and social activities). Filled-in questionnaires were returned by mail within two weeks of distribution.

## Measurements

**ICT use.**  ICT use was defined as any use of mobile devices to connect to the Internet, such as using smartphones, tablets, or wearable devices. In practice, ICT use was identified as "present" if owning at least one mobile device and "absent" if not. From mobile devices, we excluded devices that cannot be used to connect to the Internet, such as landline phones; that are not portable, such as desktop computers; and that do not work by themselves, such as Wi-Fi routers and cables.

**Activeness in life.**  The physical aspect of activeness in life was evaluated based on the expansion of the life space and the presence or absence of exercise habits. Expansion of the life space was assessed using the Life-Space Assessment (LSA) [12, 13]. The LSA examines the presence and frequency of mobility and independence across five life space levels: "home," "outside house," "neighborhood," "town," and "unrestricted." The total score ranges from 0 to 120, and higher scores indicate greater life space [13]. We divided the score on the LSA into two quartiles based on the median score. The degree of the expansion of the life space was considered to be "high" for those who scored 90 or higher and "low" for those who scored less than 90. Since the expansion of life space has been reported to be associated with sex, age, education, and medically diagnosed chronic diseases (e.g., heart and respiratory diseases) [14, 15], we used these factors as adjusted variables in this study. The presence or absence of exercise habits was assessed based on the frequency and duration of exercise and sports. According to the Japan Sports Agency's new physical fitness test implementation guidelines for individuals aged 65–79 years [16], we set the response options for the frequency of implementation as "almost every day (at least 3–4 days a week)," "sometimes (1–2 days a week)," "occasionally (1–3 days a month)," and "never." The response options for the duration were "less than 30 minutes," "30 minutes to an hour," "1–2 hours," and "2 or more hours" using the 4-subject method. Exercise habits were considered to be "present" for those who exercised at least twice a week for at least 30 minutes each time and "absent" for others. Since exercise habits have been reported to be associated with age, living alone, concerns about one's economic situation, joint pain, education level, and stroke [17], we used these factors as adjusted variables in this study.

The mental aspect of activeness in life was evaluated based on motivation. Motivation was assessed using the Japanese version of the Starkstein Apathy Scale [18, 19]. This scale comprises 14 items. A total score of 16 or more is considered to indicate the presence of apathy. Motivation was considered to be "present" for those who scored less than 16 and "absent" for those who scored 16 or higher. Since motivation has been reported to be associated with sex and age [20], we used these factors as adjusted variables in this study.

The social aspect of activeness in life was evaluated based on one's social activities. Social activities were assessed using the Scale of Social Activities among Community-Dwelling Elderly People [21]. It consists of six items: three items related to activities contributing to the community and three items related to self-enlightenment activities. The total score ranges from 6 to 18, and higher scores indicate greater participation in social activities. We divided the score on the Scale of Social Activities among Community-Dwelling Elderly People into two quartiles based on the median score. Social activities were considered to be "high" for those who scored between 12 and 18 and "low" for those who scored between 6 and 11. Since social activities have been reported to be associated with sex, living alone, education level, concerns about one's economic situation, cerebrovascular disease, and hypertension [22], we used these factors as adjusted variables in this study.

## Data analysis

We performed chi-square tests, Fisher's exact test, and the Mann-Whitney U test, as appropriate, to examine the association between ICT use and the characteristics of older adults. Logistic regression analysis was performed to estimate the odds ratios (ORs) and 95% confidence intervals (95%CIs) for the association between ICT use and outcomes. We considered sex, age, the variables associated with ICT use (p<0.05), and the variables reported to be associated with activeness in life in previous studies as potential confounders and included them in the adjustment. Other continuous variables were categorized into three levels based on the distribution of all study participants. All p-values were two-sided, and statistical significance was set at the 5% level. All statistical analyses were performed using SPSS version 28.0 (IBM Corp., Armonk, NY, USA).

## Results

The questionnaire was distributed to 4,793 participants, of whom 1,057 responded (response rate of 22.1%). We used 892 responses in the data analysis after excluding the responses of two respondents aged less than 60 years, six respondents with missing data on sex and age, and 157 respondents with missing data on ICT use (valid response rate of 18.6%).

## Participant characteristics

Table 1 presents details regarding the participants' characteristics based on their ICT use. The group that used ICT (ICT user) had a significantly higher education level than the group that did not use ICT (ICT non-user) (p < 0.001). The ICT user was significantly younger (p < 0.001) and had a significantly lower rate of concern about their economic situation (p = 0.007) than the ICT non-user. Furthermore, the ICT user had significantly lower rates of musculoskeletal disease (p = 0.023) and low back pain (p < 0.001) than the ICT non-user.

Regarding activeness in life, the mean score (± standard deviation) for the expansion of life space was 87.7 ± 22.21. The ICT user scored significantly higher than the ICT non-user (89.6 ± 20.32 versus 76.4 ± 28.86, p < 0.001). The percentage of individuals with exercise habits was significantly higher in the ICT user (57%) than in the ICT non-user (p < 0.001). The mean score (± standard deviation) for motivation was 12.5 ± 6.76. The ICT user scored significantly lower than the ICT non-user (11.9 ± 6.18 versus 16.4 ± 8.84, p < 0.001). The mean score (± standard deviation) for social activities was 11.4 ± 3.23. The ICT user scored significantly higher than the ICT non-user (11.6 ± 3.17 versus 10.2 ± 3.36, p < 0.001).

## Association between ICT use and activeness in life among community-dwelling older adults

Table 2 shows the results of the logistic regression analysis with expansion of life space, exercise habits, motivation, and social activities as dependent variables. Univariate logistic regression analysis results revealed the ORs for expansion of life space, exercise habits, motivation, and social activities in the ICT user to be 2.09 (95%CI: 1.40–3.10), 2.00 (95%CI: 1.35–2.96), 3.23 (95%CI: 2.13–4.89), and 2.14 (95%CI: 1 .44–3.19), respectively, all significantly higher than those for the ICT non-user.

In the multivariate logistic regression analysis, the variables of sex and age, those associated with ICT use (p < 0.05), and those reported to be associated with activeness in life in previous studies were included in the model. In the model that included sex, age, the variables associated with ICT use, and the variables reported to be associated with the expansion of life space in previous studies (cerebrovascular disease, heart disease, respiratory disease, diabetes

**Table 1. Participants' characteristics based on their ICT use.**

| Variable | | ICT use | | | | n = 892 | |
|---|---|---|---|---|---|---|---|
| | | absent | | present | | | |
| | | (n = 131) | | (n = 761) | | | |
| | | n | (%) | n | (%) | *p-value* | *2 |
| Sex | | | | | | | |
| | Male | 70 | (53) | 449 | (59) | 0.233 | |
| | Female | 61 | (47) | 312 | (41) | | |
| Age, years, mean ±SD | | 79.3±7.05 | | 73.6±5.54 | | p<0.001 | *3 |
| Living alone | | | | | | | |
| | Yes | 19 | (15) | 116 | (15) | 0.827 | |
| | No | 112 | (86) | 645 | (85) | | |
| Education level, years (n = 882)*1 | | | | | | | |
| | ≤9 | 32 | (26) | 48 | (06) | p<0.001 | |
| | 10–12 | 63 | (51) | 359 | (47) | | |
| | 13≤ | 29 | (23) | 351 | (46) | | |
| Concerns about one's economic situation (n = 889)*1 | | | | | | | |
| | No | 51 | (39) | 395 | (52) | 0.007 | |
| | Yes | 79 | (61) | 364 | (48) | | |
| Medically diagnosed chronic disease (n = 880)*1 | | | | | | | |
| | Cerebrovascular disease, present | 6 | (05) | 22 | (03) | 0.257 | |
| | Heart disease, present | 13 | (11) | 62 | (08) | 0.399 | |
| | Respiratory disease, present | 9 | (07) | 51 | (07) | 0.834 | |
| | Hypertension, present | 63 | (51) | 336 | (44) | 0.187 | |
| | Diabetes mellitus, present | 16 | (13) | 92 | (12) | 0.817 | |
| | Dyslipidemia, present | 10 | (08) | 106 | (14) | 0.069 | |
| | Psychiatric disease, present | 2 | (02) | 5 | (01) | 0.258 | *4 |
| | Musculoskeletal disease, present | 21 | (17) | 76 | (10) | 0.023 | |
| | Eye disease, present | 25 | (20) | 117 | (16) | 0.189 | |
| | Ear disease, present | 9 | (07) | 38 | (05) | 0.306 | |
| Subjective symptoms (n = 854)*1 | | | | | | | |
| | General fatigue, present | 12 | (10) | 62 | (09) | 0.619 | |
| | Low back pain, present | 59 | (48) | 237 | (32) | p<0.001 | |
| | Stiff shoulders, present | 34 | (28) | 193 | (26) | 0.728 | |
| | Dizziness, present | 6 | (05) | 38 | (05) | 0.899 | |
| Expansion of life space, mean ±SD (n = 858)*1 | | 76.4±28.86 | | 89.6±20.32 | | p<0.001 | *3 |
| Exercise habits (n = 838)*1 | | | | | | | |
| | absent | 73 | (60) | 310 | (43) | p<0.001 | |
| | present | 48 | (40) | 407 | (57) | | |
| Motivation, mean ±SD (n = 819)*1 | | 16.4±8.84 | | 11.9±6.18 | | p<0.001 | *3 |
| Social activities, mean ±SD (n = 843)*1 | | 10.2±3.36 | | 11.6±3.17 | | p<0.001 | *3 |

*1 Sample size differs due to missing values in questionnaire responses.

*2 The chi-square

*3 Mann-Whitney U tests

*4 Fisher's exact

**Table 2. Association between ICT use and activeness in life among community-dwelling older adults.**

| | | Expansion of life space | | | | | | | | | Exercise habits | | | | | | | |
|---|---|---|---|---|---|---|---|---|---|---|---|---|---|---|---|---|---|---|
| | | crude model (n = 858) | | | | adjusted model (n = 684) *1 | | | | | crude model (n = 838) | | | | adjusted model (n = 684) *2 | | | |
| | | n/N | (%) | ORs | 95%CI | p-value | ORs | 95%CI | p-value | n | (%) | ORs | 95%CI | p-value | ORs | 95%CI | p-value |
| ICTuse | absent | 44/123 | (36) | 1 | | | 1 | | | 48/121 | (40) | 1 | | | 1 | | |
| | present | 395/735 | (54) | 2.09 | (1.40–3.10) | p<0.001 | 1.84 | (1.03–3.28) | 0.040 | 407/717 | (57) | 2.00 | (1.35–2.96) | p<0.001 | 1.31 | (0.74–2.30) | 0.352 |

| | | Motivation | | | | | | | | | Social activities | | | | | | | |
|---|---|---|---|---|---|---|---|---|---|---|---|---|---|---|---|---|---|---|
| | | crude model (n = 819) | | | | adjusted model (n = 684) *3 | | | | | crude model (n = 843) | | | | adjusted model (n = 684) *4 | | | |
| | | n/N | (%) | ORs | 95%CI | p-value | ORs | 95%CI | p-value | n | (%) | ORs | 95%CI | p-value | ORs | 95%CI | p-value |
| ICTuse | absent | 45/107 | (42) | 1 | | | 1 | | | 43/122 | (35) | 1 | | | 1 | | |
| | present | 499/712 | (70) | 3.23 | (2.13–4.89) | p<0.001 | 2.17 | (1.22–3.84) | 0.008 | 388/721 | (54) | 2.14 | (1.44–3.19) | p<0.001 | 1.34 | (0.75–2.40) | 0.324 |

CI, confidence interval; ORs, odds ratios

* includes; variables associated with ICT use (age, education level, concerns about one's economic situation, musculoskeletal disease, low back pain), sex

*1 includes; * plus cerebrovascular disease, heart disease, respiratory disease, diabetes mellitus, psychiatric disease, ear disease, exercise habits, motivation, social activities

*2 includes; * plus lived alone, cerebrovascular disease, expansion of life space, motivation, social activities

*3 includes; * plus expansion of life space, exercise habits, social activities

*4 includes; * plus lived alone, cerebrovascular disease, hypertension, expansion of life space, exercise habits, motivation

mellitus, psychiatric disease, and ear disease), the OR for expansion of life space was significantly higher for the ICT user (OR = 1.84, 95%CI = 1.03–3.28) than in the ICT non-user. In the model that included the variables associated with ICT use and the variable reported to be associated with motivation in previous studies (sex), the OR for no apathy was significantly higher for the ICT user (OR = 2.17, 95%CI = 1.22–3.84) than for the ICT non-user. However, in the models that had exercise habits and social activities as the dependent variables, there was no association of exercise habits and social activities with ICT use.

We performed the same multivariate logistic regression analysis (results shown in Table 2) by adding personal computer (PC) users in the ICT use category (Table 3) and obtained

**Table 3. Association between ICT use, including computers and activeness in life among community-dwelling older adults.**

| | | adjusted model | | | | | | | | | | | n = 684 |
|---|---|---|---|---|---|---|---|---|---|---|---|---|---|
| | | Expansion of life space *1 | | | Exercise habits *2 | | | Motivation *3 | | | Social activities *4 | | |
| | | OR | 95%CI | p-value | OR | 95%CI | p-value | OR | 95%CI | p-value | OR | 95%CI | p-value |
| ICT use | absent | 1 | | | 1 | | | 1 | | | 1 | | |
| | present | 2.00 | (1.05–3.82) | 0.035 | 1.56 | (0.84–2.89) | 0.161 | 2.77 | (1.48–5.18) | 0.001 | 1.74 | (0.91–3.35) | 0.095 |

CI, confidence interval; OR, odds ratio

* as associated with ICT use (sex, age, education level, concerns about one's economic situation, musculoskeletal disease, low back pain)

*1 includes; * plus cerebrovascular disease, heart disease, respiratory disease, diabetes mellitus, psychiatric disease, ear disease, exercise habits, motivation, social activities

*2 includes; * plus lived alone, cerebrovascular disease, expansion of life space, motivation, social activities

*3 includes; * plus expansion of life space, exercise habits, social activities

*4 includeas; * plus lived alone, cerebrovascular disease, hypertension, expansion of life space, exercise habits, motivation

similar results. The OR for expansion of life space and no apathy was 2.00 (95%CI: 1.05–3.82) and 2.77 (95%CI: 1.48–5.18), respectively, for the ICT user and was significantly higher than the OR for the ICT non-user. In the model that had exercise habits and social activities as the dependent variables, there was no association of exercise habits and social activities with ICT use.

## Discussion

This study examined the association between ICT use and activeness in life among community-dwelling older adults. The logistic regression analysis revealed that ICT use is independently and significantly associated with the expansion of life space and no apathy, even after adjusting for confounding factors. To our knowledge, this is the first study that focuses on three aspects of older adults' activeness in life—physical, mental, and social—and examines their association with ICT use. The results suggest that ICT use can enhance activeness in life among community-dwelling older adults.

In this study, 761 participants used ICT, making up 85.3% of the total sample (n = 892). A study conducted in 2019 reported that the rate of ICT use, including the use of PCs and cell phones, is 89.4% and 82.9% among men and women, respectively, in the 70–79 years age group [23]. Since the average age of the participants in this study was 74.4 ± 6.13 years, the current results are comparable to those of the aforementioned study. While previous studies among community-dwelling older adults in Japan have reported an average score of 73.0–83.3 on the LSA [9, 24]. In this study, the average score on the LSA was 87.7, indicating that participants of this study had a greater expansion in life space than those of previous studies. This may be because the participants in this study who were registered with Silver Human Resource Centers or senior citizen clubs had many opportunities to go out for work or voluntary activities.

This study showed that the ICT user had a 1.84 times larger expansion in life space than the ICT non-user. JAGES 2016 cross-sectional study reported that older adults' recognition of the effects of ICT use, citing aspects such as increased activeness and expansion in their friendships and range of activities [5]. The results of this study are consistent with those of the previous one. In recent years, the spread of ICT has made it possible for people to easily interact with others who are far away and purchase goods and services from the comfort of their homes. As such, there is a possibility that ICT use may decrease the frequency of going out and narrow the range of activities among some people. Yet the results of this study suggest that ICT use may lead to an expansion of older adults' life space; however, the relationship between ICT use, the frequency of going out, and the expansion of life space should be examined longitudinally.

This study showed that those in the ICT user was 2.17 times more motivated than those in the ICT non-user. Additionally, the OR for motivation was the highest among the results of this study. In previous studies, a high interest in technology [25] has been reported as a factor associated with ICT use. In Japan, ICT has progressed rapidly over the past 30 years, with the fast spread of cell phones and the Internet around 1995 and smartphones since 2005 [4]. This technological innovation occurred after the participants of this study (aged 60–96 years) had already reached adulthood and old age. Consequently, contrary to younger generations, they have not been familiar with ICT devices since their childhood. Therefore, for the participants in this study, a high level of interest in such technology and a willingness to acquire new knowledge and skills may be important factors required for them to master ICT devices. ICT use has been reported to be associated with a high level of happiness [6], and it is possible that opportunities to communicate with people and obtain diverse information through ICT use

may lead to the maintenance and improvement of motivation, suggesting that ICT use may promote mental activity in particular.

In this study, exercise habits were not associated with ICT use. Previous studies conducted among adults have found that the utilization of ICT-based health management applications is associated with exercise habits [26], continued health checkups [27], and health promotion [27]. However, there was no association between exercise habits and ICT use in this study of older adults, who were less likely to use health management applications. The association between ICT use and exercise habits may be influenced by the content or platforms accessed as part of ICT use, such as applications. Furter, regarding social activities, while previous studies reported an association between active Internet use and a higher number of leisure activities among older adults [28], there was no association between ICT use and social activities in this study. This discrepancy may be explained by the fact that the participants of this study had many daily opportunities to interact with people face-to-face as part of their work and volunteer activities.

This study has two limitations. First, the participants were a specific group of individuals, those who were registered with Silver Human Resource Centers or senior citizen clubs. That is, they engage in some kind of activity, such as work and voluntary activities, in their daily life. Furthermore, they are likely to be highly active among the target groups, which might have affected the results of this study. Second, we measured only the ownership of ICT devices in the evaluation of ICT use. We did not consider the actual frequency or purpose of ICT use. Further research is needed to examine the association between specific ICT use and activeness in life.

## Conclusion

ICT use is significantly associated with expansion of the life space and motivation among community-dwelling older adults. In particularly, the highest OR value for motivation in this study suggests that ICT use may increase mental activity. These results suggest that ICT use can be a means to enhance activeness in life among community-dwelling older adults.

## Supporting information

**S1 Data.**
(XLSX)

## Acknowledgments

We thank Madoka Konishi, Associate Professor, Graduate School of Nursing, Osaka Metropolitan University, for editing a draft of this manuscript. We also thank Kazuya Ito, Statistician and Professor, Graduate School of Nursing, Osaka Metropolitan University, for critically reviewing and validating the statistical analysis conducted in this study.

## Author Contributions

**Conceptualization:** Kanna Tezuka, Yachiyo Sasaki, Midori Shirai.

**Data curation:** Kanna Tezuka.

**Formal analysis:** Kanna Tezuka, Yachiyo Sasaki.

**Funding acquisition:** Midori Shirai.

**Investigation:** Kanna Tezuka.

**Project administration:** Midori Shirai.

**Resources:** Kanna Tezuka.

**Software:** Kanna Tezuka.

**Supervision:** Yachiyo Sasaki, Midori Shirai.

**Validation:** Kanna Tezuka, Yachiyo Sasaki, Midori Shirai.

**Visualization:** Kanna Tezuka, Yachiyo Sasaki, Midori Shirai.

**Writing – original draft:** Kanna Tezuka.

**Writing – review & editing:** Kanna Tezuka, Yachiyo Sasaki, Midori Shirai.

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
