## [Decision Letter · Decision Letter 0]

2 Jul 2024

PONE-D-24-14960Association between the use of Information and Communications Technology and activeness in life among community-dwelling older adultsPLOS ONE

Dear Dr. Tezuka,

Thank you for submitting your manuscript to PLOS ONE. After careful consideration, we feel that it has merit but does not fully meet PLOS ONE’s publication criteria as it currently stands. Therefore, we invite you to submit a revised version of the manuscript that addresses the points raised during the review process. 

We look forward to receiving your revised manuscript.

Kind regards,

Tadashi Ito

Academic Editor

PLOS ONE

Additional Editor Comments (if provided):

Reviewers' comments:

Reviewer's Responses to Questions

**Comments to the Author**

1. Is the manuscript technically sound, and do the data support the conclusions?

Reviewer #1: Yes

Reviewer #2: Yes

2. Has the statistical analysis been performed appropriately and rigorously? 

Reviewer #1: Yes

Reviewer #2: I Don't Know

3. Have the authors made all data underlying the findings in their manuscript fully available?

Reviewer #1: Yes

Reviewer #2: Yes

4. Is the manuscript presented in an intelligible fashion and written in standard English?

Reviewer #1: No

Reviewer #2: Yes

5. Review Comments to the Author

Reviewer #1: The study assessed the association between the use of Information and Communications Technology (ICT) and activeness in life among community-dwelling older adults and demonstrated that ICT use can enhance the activeness of community-dwelling older adults in their lives. The findings are based on self-administered, unmarked questionnaire survey, which was conducted among individuals aged 60 or older in a specific city in Osaka. I have several comments.

1、 As shown, the questionnaire was distributed to 4,793 participants, only 1,057 responded (response rate of 22.1%), which might cause selection bias and other biases, as those who responded might have better activeness.

2、 The paper was poorly written and hard to understand. English expressions should be further refined.

3、 Table2 and 3 could be combined. And the expression of tables in the text (such as line 206-207) is not appropriate.

Reviewer #2: It is believed that this report was written with careful consideration of demographic characteristics to suit the topic.

I think this is a very well written report.

However, it is judged that the statistical aspect will require advice from experts in other statistical fields.

6. PLOS authors have the option to publish the peer review history of their article (what does this mean?). If published, this will include your full peer review and any attached files.

Reviewer #1: No

Reviewer #2: No

---

## [Author Response · Author response to Decision Letter 0]

9 Aug 2024

Dr. Tadashi Ito

Academic Editor

PLOS ONE

Dear Dr. Tadashi Ito:

We wish to re-submit the manuscript titled “Association between the use of information and communication technology and activeness in life among community-dwelling older adults.” The manuscript ID is PONE-D-24-14960.

We thank you and the reviewers for your thoughtful suggestions and insights. The manuscript has benefited from these insightful suggestions. I look forward to working with you and the reviewers to move this manuscript closer to publication in the journal PLOS ONE.

The manuscript has been rechecked and the necessary changes have been made in accordance with the reviewers’ suggestions. The responses to all comments have been prepared and are presented below. 

Thank you for your consideration of the revised paper.

Sincerely yours, 

Kanna Tezuka, RN, MSN 

Graduate School of Nursing, Osaka Metropolitan University

Osaka, Japan 

Email: so22611b@st.omu.ac.jp

PONE-D-24-14960

“Association between the use of Information and Communications Technology and activeness in life among community-dwelling older adults”

Reviewer＃１

１． As shown, the questionnaire was distributed to 4,793 participants, only 1,057 responded　

(response rate of 22.1%), which might cause selection bias and other biases, as those who responded might have better activeness.

 As the reviewer pointed out, the participants in this study engagement in activities may be relatively higher, possibly leading to selection bias. Nonetheless, this study clearly showed an association of ICT use with a high level of activeness in life (expansion of life space and motivation). Therefore, we have added a comment regarding the possibility of these biases as a study limitation in the revised manuscript (page 21, lines 334-336). 

２．The paper was poorly written and hard to understand. English expressions should be further refined.

　Addressing the reviewer’s comment, we have extensively revised the manuscript and have attempted to ensure that the rewritten manuscript is clear and comprehensible. The manuscript has also been checked by an English proofreading service and the proofreading certificate has been attached for reference. 

３．Table2 and 3 could be combined. And the expression of tables in the text (such as line 206-207) is not appropriate.

According to the reviewer’s comment, we have combined Table 2 and Table 3 in the original manuscript into Table 2 in the revised paper. In addition, we have inserted Tables 1 and 2 immediately after the first paragraph in which they are cited according to the submission guidelines and confirmed that the tables are accurately formatted with appropriate captions in the revised manuscript. 

PONE-D-24-14960

“Association between the use of Information and Communications Technology and activeness in life among community-dwelling older adults”

Reviewer＃2

It is believed that this report was written with careful consideration of demographic characteristics to suit the topic. I think this is a very well written report. However, it is judged that the statistical aspect will require advice from experts in other statistical fields.

According to the reviewer’s comment, we have asked an expert in statistics, Professor Kazuya Ito, Statistician, Graduate School of Nursing, Osaka Metropolitan University, to evaluate the statistical analysis conducted in this study. Consequently, the statistical analysis procedures were critically reviewed and validated by him. We have added a comment regarding this in the revised manuscript (page 22, lines 356-358) and attached the certificate of statistical analysis.

---

## [Decision Letter · Decision Letter 1]

16 Sep 2024

Association between the use of Information and Communication Technology and activeness in life among community-dwelling older adults

PONE-D-24-14960R1

Dear Dr. Kanna Tezuka,

We’re pleased to inform you that your manuscript has been judged scientifically suitable for publication and will be formally accepted for publication once it meets all outstanding technical requirements.

Kind regards,

Tadashi Ito

Academic Editor

PLOS ONE

Reviewers' comments:

Reviewer's Responses to Questions

**Comments to the Author**

1. If the authors have adequately addressed your comments raised in a previous round of review and you feel that this manuscript is now acceptable for publication, you may indicate that here to bypass the “Comments to the Author” section, enter your conflict of interest statement in the “Confidential to Editor” section, and submit your "Accept" recommendation.

Reviewer #1: All comments have been addressed

2. Is the manuscript technically sound, and do the data support the conclusions?

Reviewer #1: Yes

3. Has the statistical analysis been performed appropriately and rigorously? 

Reviewer #1: Yes

4. Have the authors made all data underlying the findings in their manuscript fully available?

Reviewer #1: Yes

5. Is the manuscript presented in an intelligible fashion and written in standard English?

Reviewer #1: Yes

6. Review Comments to the Author

Reviewer #1: No more comments.Please use the space provided to explain your answers to the questions above. You may also include additional comments for the author, including concerns about dual publication, research ethics, or publication ethics. (Please upload your review as an attachment if it exceeds 20,000 characters) (Limit 100 to 20000 Characters)

7. PLOS authors have the option to publish the peer review history of their article (what does this mean?). If published, this will include your full peer review and any attached files.

Reviewer #1: No

---

## [Editor Report · Acceptance letter]

19 Sep 2024

PONE-D-24-14960R1 

PLOS ONE

Dear Dr. Tezuka, 

I'm pleased to inform you that your manuscript has been deemed suitable for publication in PLOS ONE. Congratulations! Your manuscript is now being handed over to our production team.

Kind regards, 

on behalf of

Dr. Tadashi Ito 

Academic Editor

PLOS ONE